# Delivery of Nucleotide Sugars to the Mammalian Golgi: A Very Well (un)Explained Story

**DOI:** 10.3390/ijms23158648

**Published:** 2022-08-03

**Authors:** Dorota Maszczak-Seneczko, Maciej Wiktor, Edyta Skurska, Wojciech Wiertelak, Mariusz Olczak

**Affiliations:** Faculty of Biotechnology, University of Wroclaw, Joliot Curie 14A, 50-383 Wrocław, Poland; dorota.maszczak-seneczko@uwr.edu.pl (D.M.-S.); maciej.wiktor@uwr.edu.pl (M.W.); edyta.skurska2@uwr.edu.pl (E.S.); wojciech.wiertelak@uwr.edu.pl (W.W.)

**Keywords:** nucleotide sugar transporter, glycosylation, SLC35A1, SLC35A2, SLC35A3, SLC35C1, SLC35B4

## Abstract

Nucleotide sugars (NSs) serve as substrates for glycosylation reactions. The majority of these compounds are synthesized in the cytoplasm, whereas glycosylation occurs in the endoplasmic reticulum (ER) and Golgi lumens, where catalytic domains of glycosyltransferases (GTs) are located. Therefore, translocation of NS across the organelle membranes is a prerequisite. This process is thought to be mediated by a group of multi-transmembrane proteins from the SLC35 family, i.e., nucleotide sugar transporters (NSTs). Despite many years of research, some uncertainties/inconsistencies related with the mechanisms of NS transport and the substrate specificities of NSTs remain. Here we present a comprehensive review of the NS import into the mammalian Golgi, which consists of three major parts. In the first part, we provide a historical view of the experimental approaches used to study NS transport and evaluate the most important achievements. The second part summarizes various aspects of knowledge concerning NSTs, ranging from subcellular localization up to the pathologies related with their defective function. In the third part, we present the outcomes of our research performed using mammalian cell-based models and discuss its relevance in relation to the general context.

## 1. Introduction to Glycosylation

Glycosylation is a process of enzymatic attachment of sugar residues to macromolecules [1]. This process occurs due to the sequential action of glycosyltransferases [2]. Glycosylation takes place mainly inside the endoplasmic reticulum (ER) and the Golgi apparatus of eukaryotic cells. The phenomenon of cytoplasmic and nuclear glycosylation has also been described [3] and involves the attachment of a single *N*-acetylglucosamine (GlcNAc) residue to proteins (protein O-GlcNAcylation).

The main classes of glycoconjugates are distinguished by the type of macromolecule to which carbohydrate moieties are attached and include glycoproteins, proteoglycans, and glycolipids. Glycoproteins are proteins with one or more covalently attached oligosaccharide chains; proteoglycans are proteins with one or more covalently linked glycosaminoglycan (GAG) chains; glycolipids are lipids with one covalently attached oligosaccharide chain.

In glycoproteins, carbohydrate moieties, i.e., glycans, can be attached to a polypeptide chain in two main ways. N-glycans are attached to the amide nitrogen of an asparagine residue found within the characteristic amino acid triplet Asn-X-Ser/Thr (where X stands for any amino acid residue except proline), while O-glycans are attached to the oxygen atom of serine or threonine.

N-glycan biosynthesis begins on the cytoplasmic side of the ER membrane with the formation of an oligosaccharide precursor composed of fourteen sugar residues (GlcNAc_2_-Man_9_-Glc_3_) attached to dolichol phosphate [4]. Dolichol phosphate enables the transfer of the oligosaccharide precursor to the ER lumen, where it is subsequently transferred onto the asparagine residue of the nascent polypeptide chain. Several further modifications in the ER and Golgi apparatus, made by the appropriate glycosidases and glycosyltransferases, lead to the formation of mature N-glycans. Based on the type of predominating sugars, three types of N-glycans, i.e., high mannose (Man), complex and hybrid, can be distinguished, all of which share a common pentasaccharide (Man_3_-GlcNAc_2_) core.

Biosynthesis of mucin-type O-glycans begins with the addition of an N-acetylgalactosamine (GalNAc) to a serine or threonine residue of the protein [5]. This reaction takes place in the Golgi apparatus [6]. As a result, the simplest mucin-type O-glycan is formed which can be further extended, eventually leading to the formation of several types of core structures composed of a few to a dozen or so monosaccharides. In addition to GalNAc, typical mucin-type O-glycans contain galactose (Gal), GlcNAc, fucose (Fuc), and sialic acid (Sia).

Proteoglycans are composed of core proteins and GAG chains. Most GAGs are linear polysaccharides made of several repeats of disaccharide units formed by hexosamine and uronic acid. The polymerization of the GAG chains takes place in the Golgi apparatus, where they subsequently undergo several modifications including sulfation. 

Glycosphingolipids (GSLs) are glycolipids composed of a ceramide and an oligosaccharide. Glycosphingolipids are essential components of the plasma membranes of mammalian cells. The precursor of the majority of GSL species is glucosylceramide (GlcCer) which is synthesized on the cytoplasmic side of the Golgi membrane [7]. Glucosylceramide can be subsequently extended in the Golgi lumen by a series of glycosyltransferases. The resulting GSLs may contain Gal, GalNAc, GlcNAc and Sia residues. A small subset of GSLs is derived from galactosylceramide (GalCer) which is synthesized in the ER lumen [8].

The biosynthesis of glycoconjugates is mediated by glycosyltransferases. These are usually type II membrane proteins with a short N-terminal cytoplasmic tail, a single transmembrane domain (TMD) followed by a so-called stem region and a C-terminal catalytic domain located in the ER/Golgi lumen [9]. However, soluble as well as multi-transmembrane glycosyltransferases have also been identified [10,11].

Glycosylation plays many fundamental roles, especially in higher multicellular organisms. It mediates a wide range of biological recognition events of both intrinsic (e.g., interactions of selectins with their ligands during the leukocyte adhesion cascade) and extrinsic (e.g., binding and adhering of pathogens to host cells) origin. Moreover, it regulates signal transduction pathways and intracellular trafficking events. In many cases, the presence of the carbohydrate moiety increases stability of the modified protein (glycosylated proteins are usually more resistant to denaturing factors and proteolysis). 

The donors of sugar residues for the majority of glycosylation reactions are activated forms of monosaccharides, i.e., nucleotide sugars (NSs). Activation of monosaccharides involves the conversion of a respective sugar with either a mono- (CMP) or diphosphonucleotide (GDP/ UDP). Nucleotide sugars are synthesized in mammalian cells and their commonly used abbreviations are listed in Table 1.

Activation of the majority of monosaccharides takes place in the cytoplasm [12]. The exception is CMP-Sia, the biosynthesis of which takes place in the nucleus [13]. Nucleotide sugars are relatively large (~550–650 Da), hydrophilic and double negatively charged molecules. Therefore, they cannot diffuse freely across the membranes of organelles where glycosylation takes place, which requires the existence of specific carriers in the ER and Golgi membranes. This role is thought to be played by multi-transmembrane proteins from the solute carrier 35 (SLC35) family, i.e., nucleotide sugar transporters (NSTs). 

As already mentioned, the vast majority of glycosylation-related events take place in the ER/Golgi lumens. However, a subset of glycosylation reactions occurs in the cytoplasm with no requirement for NS transport across the organelle membranes. One of such examples is the O-GlcNAc modification of cytoplasmic and nuclear proteins [3]. In addition, GlcCer is synthesized on the cytoplasmic side of the Golgi membrane with the involvement of UDP-glucose ceramide glucosyltransferase (UGCG) [7]. Finally, the first stages of N-glycan assembly, i.e., the synthesis of the so-called lipid-linked oligosaccharide, occur on the cytoplasmic side of the ER membrane [4]. Thus, it must be kept in mind that not all glycosylation reactions require translocation of NS across the organelle membranes.

## 2. Methodological Approaches to the Nucleotide Sugar Transport

### 2.1. Glycosylation-Deficient Mutants

The way to understand the biology of glycosylation has been long and bumpy. In 1970s, when the family of NSTs had yet to be identified, two groups working with Chinese hamster ovary (CHO) cells reported isolation of several clones resistant to the toxic effects of plant lectins.

An important piece of research was accomplished by Stanley et al. who isolated eight different clones (referred to as CHO-Lec^R^ lines) for their ability to thrive in the presence of lethal concentrations of phytohemagglutinin, wheat germ agglutinin (WGA), ricin or the agglutinin(s) from *Lens culinaris* [14,15]. Subsequently, as most of these mutants behaved recessively in somatic cell hybrids, the CHO-Lec^R^ phenotypes could be classified into seven complementation groups, I-VII [15]. To increase the rate of genetic alterations, the parental CHO cell line was initially pre-treated with a mutagen, ethylmethanesulfonate, prior to seeding onto lectin-containing media for selection [14], but later it was demonstrated that mutations leading to the same phenotypes occur in unmutagenized CHO populations spontaneously [16]. Over the next few years, additional CHO-Lec^R^ phenotypes and complementation groups joined the pool [16,17].

Independently of the above, and without usage of any mutagenic factors, Briles et al. selected three clones for their ability to tolerate lethal dosages of WGA [18]. Further experiments led to the observations, that in comparison to the parental cell line, two of the surviving clones, i.e., clones 1 and 1021, exhibited increased binding of *Ricinus communis* agglutinin I (RCA), ricin and soybean agglutinin, whereas clone 13 showed a decreased binding of RCA. Subsequent analysis of the total membrane carbohydrate composition revealed that clones 1 and 1021 had a selective reduction of Sia content, while clone 13 was deficient in both Sia and Gal, which rationalized the altered lectin binding. Although, the mutations were not directly associated with non-functional NSTs, the mutated clones were unable to perform glycosylation reactions despite the presence of appropriate glycosyl donors, acceptors, and glycosyltransferases.

In parallel to the work in CHO cells, glycosylation-impaired mutants were also found in yeast. Smith et al. reported two clones of *Kluyveromyces lactis* with GlcNAc-deficient cell surface determinants, one of which, referred to as mnn2-2, was defective in attaching GlcNAc to polymannose [19]. The mutant was initially interpreted to have compromised glycosyltransferase membrane orientation or defective substrate compartmentation [20], however, it was later proven that mnn2-2 had impaired transport of UDP-GlcNAc [21].

Glycosylation mutants were also generated in *Schizosaccharomyces pombe* [22]. Here, the parental strain was exposed to ethylmethanesulfonate mutagen, recovered overnight, and agglutinated with Gal-specific peanut (*Arachis hypogaea*) lectin. Cells with deficiency in cell surface galactosylation remained in the supernatant after centrifugation and were subsequently selected based on staining with peroxidase-conjugated lectins. The *gms1* mutant that was generated had over 6-fold reduced Man:Gal ratio in N-linked polysaccharides. The Gal-deficient mutation rendered *gms1* mutant vanadate-sensitive and allowed a convenient screening for complementing genes using the *S. pombe* genomic library [23]. Sequencing revealed that all four vanadate-resistant transformants harboured an open reading frame (ORF) with 63% similarity to the human UDP-Gal transporter, later known as SLC35A2.

Further, glycosylation-deficient mutants were also generated in *Saccharomyces cerevisiae* [24]. By spreading haploid yeast cultures onto plates containing sodium orthovanadate and allowing the growth for several days, several spontaneous vanadate-resistant clones were obtained. The mutants, referred to as Vrg, were all associated with glycosylation defects as demonstrated by changes in gel mobility of external invertase protein. It was later shown that one of the mutated genes, i.e., *vrg4*, is required for the GDP-Man uptake and it is homologous to the aforementioned *K. lactis mnn2* and other putative NST-encoding genes [25]. 

In an alternative approach, mouse mammary carcinoma FM3A cells were mutagenized with N-methyl-N’-nitro-N-nitrosoguanidine, recovered and infected with Newcastle disease virus [26]. Cells prone to the infection displayed hemagglutinin-neuraminidase glycoproteins and stained positive in hemadsorption test. The mutant, Newcastle disease virus-resistant colonies were significantly less hemadsorptive and were shown to produce incomplete N-linked glycans and to have altered lectin cross-reactivity like that of the CHO-Lec8 mutant. Increased sensitivity to a *Griffonia simplicifolia* lectin, GS-II, is characteristic for GlcNAc exposure, which suggested that the mutants, designated as Had-1, were incapable of transferring Gal.

Aside from CHO cells, yeast and mouse, lectin-resistant mutants were also generated in Madin–Darby canine kidney cells (MDCK). Following pre-incubation with N-methyl-N’-nitro-N-nitrosoguanidine mutagen, parental MDCK cells were selected for resistance to concanavalin A and RCA [27]. The *R. communis* agglutinin-resistant mutant, MDCK-RCA^R^, was shown to expose surface glycoproteins and glycolipids affected by Gal deficiency and to resemble CHO clone 13 in its properties [28].

In summary, thanks to several accompanying side-effects such as altered lectin sensitivity, increased vanadate/viral resistance, etc., glycosylation-deficient mutants were isolated in multiple different organisms. Although in most of the studies a direct link to a particular transport mechanism was not clearly established, the effects have been associated with a deficiency in nucleotide sugar transport based on variety of indirect evidence such as: (i) unaffected glycosyltransferase activity, (ii) insufficient translocation of NSs in the mutant-derived microsomes, (iii) restoration of the wild-type phenotype by transformation with a sequence encoding for homo- or heterologous (putative) NST, (iv) sequence homology to the (putative) NST gene from another organism, etc. Mutants with impaired glycosylation prepared the ground and became an invaluable tool for identifying and studying NSTs’ functions.

### 2.2. Functional Genetic Complementation

Glycosylation-deficient mutants created by spontaneous or induced mutagenesis do not identify the specific genetic defects that underly the altered phenotypes. Hence, to link the observed deficiencies and the affected gene(s), a strategy of genetic complementation was used. In this method glycosylation mutant is transformed with a wild-type genomic library and selected for the wild-type phenotype features (e.g., antibody or lectin binding). Subsequently, the genetic fragments that corrected (complemented) the mutation are sequenced and mapped to specific ORFs.

In this way the *Leishmania donovani* C3P0 clone with a defect in lipophosphoglycan (LPG) synthesis was found to harbour a deletion in a gene termed *Lpg2* encoding an ORF homologous to yeast *Vrg4* [29]. Using the same approach, the *K. lactis MNN2-2* ORF was demonstrated to encode a multi-transmembrane protein, which corrected the ability to transport UDP-GlcNAc into Golgi-enriched vesicles [30].

Genetic complementation can also be achieved not by transformation but because of the fusion of two cells, one of which is devoid of glycosylation while the other brings a functional copy of the responsible gene. In this way it is possible to check whether two glycosylation-deficient mutants of unknown genotype are defective with respect to the same (lack of complementation) or different (positive complementation) genes. In the case of mammalian cells, fusion is typically induced by incubation of both species in the presence of polyethylene glycol.

In this manner ethylmethanesulfonate was used to generate the CHO-6B2 mutant that was found via cell fusion experiments to belong to the CHO-Lec2 complementation group [31]. The mutant was complemented with a murine cDNA library and the resulting ORF encoded a multi-transmembrane protein, by sequence analysis, that localized to the Golgi and showed high sequence similarity to the plant ammonium transporters.

A complementation approach was also used for more evolutionarily distant pairs of organisms. Two isoforms of a putative human UDP-Gal transporter (UGT1 and UGT2) were identified by complementation of the genetic defect of murine Had-1 cells by human cDNA library [32,33]. The ORFs identified in this way were able to reverse Had-1 lectin-resistance spectrum to that of the parental FM3A cells.

Phenotypic complementation was also reported between kingdoms. Yeast genes were able to correct NS transport deficiencies in canine or hamster cell lines. The *Kluyveromyces lactis mnn2-2* mutant regained ability to bind GS-II lectin when transformed with cDNA library of MDCK cells even though the putative mammalian UDP-GlcNAc transporter shared very little amino acid sequence identity to its yeast kin [34].

A gene, referred later as *gms1*^+^, was amplified from *S. pombe* genome and was shown to restore cell wall galactosylation of the *gms1* mutant [23]. The product of this gene was also demonstrated to complement CHO-Lec8 cells deficient in UDP-Gal transport activity [35]. Similarly, putative human CMP-Sia could complement CHO-Lec2 as demonstrated by a restored lectin sensitivity pattern and NS transport into microsomal vesicles [36].

The multi-transmembrane protein Sqv-7 from *Caenorhabditis elegans* was shown to complement the MDCK-RCA^r^ phenotype, suggesting that heterologous over-expression of Sqv-7 can restore UDP-Gal translocation into the canine Golgi.

Nematode sequences homologous to known NSTs were able to complement glycoconjugate fucosylation impaired in the LADII patient fibroblasts [37]. As no mutations were found in LADII fucosyltransferases, it suggested a defect in NS translocation. Subsequently, based on sequence homology, a putative GDP-Fuc transporter was identified in *Drosophila melanogaster* [38]. The LADII cells complemented with *D. melanogaster CG9620* cDNA showed a partial reversion to the wild-type fucosylation phenotype as demonstrated by *Pisum sativum* agglutinin staining of the cell lysates.

More examples of the applications of functional genetic correction in the NST studies include complementation of: (i) CHO-Lec8 with *D. melanogaster* UDP-Gal/UDP-GalNAc transporter [39], (ii) *S. cerevisiae* vrg4 mutant with a *Candida glabrata* homologue, termed CgVRG4 [40], (iii) CHO-Lec2 with human CMP-Sia transporter sequence, SLC35A1 [41], (iv) CHO-Lec8 with two isoforms of putative UDP-Gal transporter from *Arabidopsis thaliana* [42], and others.

In summary, functional genetic complementation has been widely used to demonstrate the ability of an unknown gene to substitute for a certain deficiency of a previously identified glycosylation-impaired mutant and using this approach multiple NSTs from different organisms were identified. Although a positive genetic complementation with respect to a single phenotypic parameter is not yet sufficient to comprehensively define function of a newly studied ORF, it is certainly a good starting point for further experiments such as electrophoretic mobility analysis of cell surface glycoconjugates, lectin staining/sensitivity panel or vesicle-based in vitro transport assay. The robustness of the method seems to be due the extraordinary versatility of the NSTs, which function appears essentially insensitive to changes in their molecular environments across different species and kingdoms.

### 2.3. Microsome Penetration Assays

In parallel to the phenotypic observations made using glycosylation-deficient mutant cells, NS transport was investigated more directly using in vitro transport/vesicle penetration assays. The first speculation about a specific transport of NSs into the Golgi lumen dates to 1975 when Kuhn and White showed evidence that Golgi vesicles exhibited selective permeability towards different UDP-sugars [43].

A few years later, the radioactivity-based approach adopted by Carey et al. allowed direct monitoring of the fate of the ^3^H- or ^14^C-labeled NS substrates added to the tissue-derived Golgi microsomes [44]. In this method microsomes were isolated by ultracentrifugation of the rat liver homogenate supernatant and incubated in the presence of radiolabelled CMP-Sia. The reaction was stopped by dilution, vesicles were recovered by centrifugation, and the transport could be calculated, as the total solutes, and those outside and between vesicles in the pellet were known. The study showed accumulation of CMP-Sia inside the Golgi vesicles. The NS transport was also reported in this way for GDP-Fuc [45], adenosine 3′-phosphate 5′-phosphosulfate (PAPS) [46], and for the UDP-Glc, the latter into the ER lumen of rat liver cells [47].

An analogous approach was applied to measure NS transport in other organisms. In *S. cerevisiae*, over 40-fold accumulations of GDP-Man [48] and 22-fold accumulations of UDP-Glc [49], as compared to the incubation medium, were reported. A differential translocation of GDP-Fuc was observed in human fibroblasts from leukocyte adhesion deficiency II (LADII) patient as compared to the control fibroblasts from healthy individuals [50].

A simpler approach to study translocation of NS into microsomes was presented by Waldman and Rudnick who reported an over 20-fold concentration of UDP-GlcNAc inside rat liver Golgi vesicles [51]. Here, after the incubation of microsomal vesicles with radiolabelled substrates, samples were poured onto filters and the non-translocated NS was removed by washing. The difference of this method with respect to the pioneer approach of Perez and Hirschberg [52] was that the total transport activity was approximated by the total radioactivity in the washed vesicle pellet that remained on the filters. This approach was also applied to show translocation of UDP-GlcNAc [53] and UDP-GlcA [54] into ER membranes of rat liver cells and GDP-Man uptake in *L. donovani* microsomal vesicles [55].

In summary, microsome penetration experiments did not reveal identities of specific proteins capable of translocation of NSs across the Golgi/ER membranes but proved the existence of carrier-mediated systems for which substrate specificity, K_m_, V_max_, temperature/inhibitor dependence, sensitivity to proteases/detergents etc. could be measured.

### 2.4. Heterologous Expression

Upon identification and cloning of the genes required for translocation of NSs, it became possible to over-express these genes and study the function of the resulting protein products in heterologous systems. Such a concept was not free of disadvantages but because different organisms produce different glycans, it offered possibility to use a system with a satisfactory low transport background with respect to a specific NS.

Murine CMP-Sia transporter was over-expressed in *S. cerevisiae* because, contrary to mammals, yeast do not sialylate their glycoconjugates, hence should not possess CMP-Sia transporters in the Golgi membranes [56]. The over-expression of the gene was triggered with 2% Gal and the vesicles from the induced cells became able to transport CMP-Sia (9-fold enrichment). Unexpectedly, induction with Gal also increased ability to transport UDP-Gal (2.3-fold enrichment), even though normally Gal is not incorporated into yeast cell wall proteins. The effect of Gal in culture medium on UDP-Gal transport abilities was later described by Tiralongo et al. [57].

The long list of human NSTs that have been over-expressed and studied in *S. cerevisiae* include transporters of UDP-Gal (SLC35A2) [58], UDP-GlcNAc (SLC35A3) [59], bispecific UDP-GlcA/UDP-GalNAc (UGTrel7/SLC35D1) [60], UDP-GlcNAc/UDP-Glc (HFRC1/SLC35D2) [61,62] and UDP-Xyl/UDP-GlcNAc (SLC35B4) transporters [63].

Genes of human UDP-Gal transporters (both UGT1 and UGT2) were over-expressed under strong constitutive yeast GAPDH promoter and the membrane vesicles were able to transport radioactive UDP-[^3^H]-Gal but not CMP-[^3^H]-Sia [58]. Here, however, the effect of Gal in the medium was not controlled for and the specificity towards other solutes was not investigated. The same group also analysed transport after copper-induced over-expression of the putative human UDP-GlcNAc transporter [59]. In this case a specific increase in UDP-GlcNAc transport was observed in the absence of CMP-Sia and UDP-Gal translocation but this time Gal was omitted from the medium. Substrate specificity was most widely investigated for a candidate NST gene (referred to as hUGTrel7/SLC35D1) where eight different NSs were tested and selective enrichments were observed for UDP-GlcA and UDP-GalNAc, and maybe for UDP-GlcNAc [60]. A wide panel of substrates was also tested for SLC35B4, which showed dual specificity for UDP-Xyl and UDP-GlcNAc in the absence of translocation of UDP-Gal, UDP-GalNAc, UDP-GlcA and GDP-Fuc [63]. In the case of the second UDP-GlcNAc transporter (SLC35D2), some specificity was also observed towards UDP-Glc [62].

The yeast system was also used to study putative *C. elegans* NST genes. Golgi vesicles from *S. cerevisiae* over-expressing SQV-7 concentrated UDP-GlcA, UDP-Gal and UDP-GalNAc inside the lumen but not CMP-Sia, GDP-Fuc, UDP-GlcNAc [64]. The vesicles with over-expressed SFR-3 translocated UDP-Gal and UDP-GlcNAc [65] and those with over-expressed *C. elegans C03H5.2* accumulated UDP-GlcNAc and UDP-GalNAc [66]. There is also a case of a multi-specific transporter encoded by the gene *ZK896.9* [67] which was shown to transport UDP-Glc, UDP-Gal, UDP-GlcNAc and UDP-GalNAc.

The nucleotide sugar transporters of *D. melanogaster* were heterologously over-expressed in yeast too. Segawa et al. showed that the UDP-Gal transporter over-expressed from a copper-inducible vector had dual specificity towards UDP-Gal and UDP-GalNAc and did not transport UDP-Glc, UDP-GlcNAc, UDP-GlcA, UDP-Xyl nor CMP-Sia [39].

Aside from *S. cerevisiae* a heterologous expression of an NST was reported in *Pichia pastoris* [68]. The murine transporter of CMP-Sia was over-expressed under a strong methanol-induced promoter and showed functionality in phosphatidylcholine (PC) proteoliposomes when solubilized and IMAC-purified in *n*-nonyl-*β*-d-maltopyranoside. Its activity was inhibited by CMP, DIDS and was maintained in only a single detergent.

Finally, NSTs were also over-expressed in *Escherichia coli* with an intention to produce the protein in quantities sufficient for structural studies. Initial trials with murine CMP-Sia transporter resulted in over-expression into inclusion bodies and the protein required solubilization in 8 M urea and refolding [57]. However, after subsequent incorporation into artificial PC-liposomes specific transport of CMP-Sia into the lumen of the vesicles was observed. None of the tested UDP-NSs could translocate and the transport was inhibited by CMP and Triton X-100.

To avoid insoluble expression, the murine CMP-Sia transporter was N-terminally fused to the OmpA signal sequence and targeted to the *E. coli* inner membrane [69]. An ability to transport CMP-Sia was exhibited by both spheroplasted *E. coli* and mixed PC-*E. coli* inner membrane proteoliposomes. An attempt to produce functional NST was also made for human GDP-Fuc transporter (SLC35C1) [70]. The prepared inside-out *E. coli* membrane vesicles achieved up to 40-fold enrichment of GDP-Fuc when bacteria were transformed with OmpA-GDP-Fuc transporter fusion protein.

In summary, heterologous expression has clearly boosted functional studies of NSTs allowing the identification of novel types of transporters and establishing their substrate specificities. Examples of functional expression in yeast and *E. coli* also proved that these proteins can remain active in lower organisms. This in turn has paved the way for development and refinement of purification protocols which provided pure protein samples for the upcoming structural studies. Among multiple advances that have been brought to the field through these studies a few uncertainties remain as, for example, to what extent the characteristics of the heterologously-expressed NST can be influenced by the molecular environments of the heterologous hosts. 

### 2.5. Reconstitution into Liposomes

The attempts to reconstitute NS transport activity into artificial membrane systems were undertaken already in 1980s, when it was demonstrated that CMP-Sia and PAPS transport activities are maintained in PC-liposomes [71]. The Golgi protein extract from rat liver was mixed with PC-liposomes and the five step freeze-thawing procedure was used to reconstitute protein into the liposomes. The obtained mixture was sonicated and purified using size exclusion chromatography. The characteristics of the CMP-Sia and PAPS transport in proteoliposomes and in the intact Golgi vesicles turned out similar. In the same manner UDP-Gal, UDP-Xyl and UDP-GlcA transport activities were successfully reconstituted, however, characteristics of UDP-Xyl transport in PC liposomes were perturbed (little temperature dependence, insensitivity to the inhibitor) as compared to those in Golgi lipids [72].

Further advancement in the method came with the work of Puglielli et al. who were able to purify rat UDP-GalNAc transporter to apparent homogeneity using conventional multi-step column chromatography [73]. The protein was incorporated into PC-liposomes (by freeze-thawing) to monitor the activity of the transporter at various purifications steps and appeared functional. The achievement was repeated with GDP-Fuc transporter [74], which suggests that rat NSTs tolerate extraction with Triton X-100 and that depletion of cellular lipids does not jeopardize their function.

A somewhat different approach was presented for the case of the *Leishmania donovani* LPG2 GDP-Man transporter [75]. The protein’s C-terminus was fused to a hexahistidine tag and over-expressed homologously in a *L. donovani lpg2*−/− knockout. In this case, however, LPG2 could not be efficiently extracted with Triton X-100 and glycodeoxycholic acid was used instead. Solubilized LPG2 was purified using IMAC and reconstituted into PC-liposomes using freeze-thawing (supported by subsequent detergent depletion with SM-2 polystyrene beads). Proteoliposomes prepared in this way incorporated radioactive GDP-Man, GDP-arabinose (GDP-Ara) and GDP-Fuc but not UDP-Gal.

Reconstitution into artificial membranes was also used as a quality-check procedure in the aforementioned case of the over-expression of the murine CMP-Sia transporter in *P. pastoris* [68] and *E. coli* [57]. Here, proteoliposomes were formed either by conventional freeze-thawing method [57] or by mixing of PC-liposomes pre-incubated with a detergent with the protein purified in the same detergent; detergent was subsequently removed with SM2 polystyrene beads [68].

In summary, reconstitution of NSTs into artificial membranes offered several useful possibilities in NST studies. Liposomes enable studies of the function of putative transporter proteins for which substrate specificity is unknown. They ensure a clear background system to study kinetics parameters for cases, where finding a zero background host or generation of genes knockout turns problematic. Proteoliposomes seem also a first-choice option in the case of NST redundancy, i.e., a presence of multiple proteins capable of transporting the same substrate. Finally, due to its defined molecular character, liposomes provide means of understanding regulatory factors that affect NS transport. On the other hand, proteoliposomes are still an artificial membrane environment and some discrepancies between the characteristics of the reconstituted NSTs and those in their native molecular context may occur.

## 3. Nucleotide Sugar Transporters

### 3.1. Current State of Structural Studies on NSTs

The identification of genes encoding nucleotide sugar transporters from different species, initiated in the mid-1990s, revealed the existence of a group of related proteins with a high degree of amino acid sequence similarity [76]. The degree of this similarity does not seem to correlate with the substrate specificity of the transporters, as the amino acid sequences of NSTs with the same specificity coming from different species are less like each other than the amino acid sequences of NSTs with different specificities derived from the same species.

Nucleotide sugar transporters are relatively small proteins, composed of 320–400 amino acid residues. Hydrophobicity profiles of the amino acid sequences of NSTs allow to classify them as type III membrane proteins with 6–10 TMDs. Detailed topology was first experimentally determined for the mouse CMP-Sia transporter (CST) [77]. In this study, CST was shown to contain 10 TMDs with N- and C-termini facing the cytoplasm.

A major breakthrough in the studies on the NST structure was made by the crystallization of the selected NSTs. In 2019, a 3-D structure of the yeast GDP-Man transporter (Vrg4), was obtained [78]. In the same year, the mouse CMP-Sia transporter was crystallized [79]. The results of these studies revealed the arrangement of TMDs, allowed for the characterization of substrate binding sites and, in the case of Vrg4, revealed the requirement for short-chain lipids in the membrane environment. Vrg4 was more active in the short- than in the long-chain lipid environment (short-chain lipids were hypothesized to enable conformational changes of Vrg4 required for the transport to occur).

### 3.2. Subcellular Localization of NSTs

All the NSTs identified to date are located either in the Golgi apparatus or in the endoplasmic reticulum. So far, the only NST displaying a dual localization is one of the SLC35A2 splicing variants, UGT2 [80]. The intracellular distribution of at least some of these proteins is determined by the presence of certain specific sequence motifs. The localization of UGT2 in the ER is determined by the C-terminal motif KVKGS [80]. The presence of similar sequences (e.g., KKTSH in SLC35B1, KDSKKN in SLC35B4, KGKGAV in SLC35D1) causes the membrane proteins to be retained in the ER [81].

The murine CMP-Sia transporter is located in the Golgi apparatus due to the presence of the C-terminal IIGV motif, as deletion of this sequence resulted in the retention of the NST in the ER membrane [82]. The C-terminal valine residue was shown to serve as an export signal from the ER [83]. In the case of the yeast GDP-Man, transporter amino acids 16–44 present in the N-terminal domain were shown to be a determinant of the correct subcellular localization [84].

Some data suggest that the intracellular distribution of NSTs is affected by their interactions with other membrane proteins. The subcellular localization of the Golgi-resident variant of the UDP-Gal transporter (UGT1) changed upon the association with the galactosylceramide synthase (UGT8) [82]. When over-expressed separately, UGT1 localized to the Golgi complex [80,85] whereas UGT8 was found in the ER [8]. Upon the simultaneous over-expression of UGT1 and UGT8, the former was shown to localize to the ER [86].

### 3.3. Substrate Specificity of NSTs

For many years NSTs were considered monospecific [87]. According to this assumption, each transporter would be responsible for the specific transfer of only one type of NS into the ER/Golgi lumen. Specificity towards more than one NS was demonstrated for some NSTs identified in lower organisms such as *Leishmania* sp. [88]. One of NSTs from *Leishmania*, LPG2, was shown to transport GDP-Man, GDP-Ara and GDP-Fuc [55].

Studies performed on *C. elegans* provided more examples of multi-specific NSTs. In the genome of *C. elegans*, 18 sequences coding for potential NSTs were identified, while the glycoconjugates produced by this nematode consist of only seven types of monosaccharides [67]. The SQV-7 transporter from *C. elegans* showed specificity for UDP-Gal, UDP-GlcA and UDP-GalNAc [64]. The *srf-3* gene encodes a nematode membrane transporter specific for UDP-GlcNAc and UDP-Gal [65]. The protein encoded by the *C. elegans CO3H5.2* gene was shown to be specific for UDP-GlcNAc and UDP-GalNAc [66]. The ZK896.9 transporter showed specificity for UDP-Glc, UDP-GlcNAc, UDP-GalNAc and UDP-Gal [67].

Multi-specific transporters were also identified in humans. An example is the UGTrel7/SLC35D1 protein specific for UDP-GlcNAc, UDP-GalNAc and UDP-GlcA [60,89]. Segawa et al. showed that the human UDP-Gal transporter as well as its homologue from *D. melanogaster* can also translocate UDP-GalNAc [39]. The Frc transporter identified in the fruit fly was shown to be specific for UDP-GlcA, UDP-GlcNAc, UDP-Xyl [90], UDP-Gal, UDP-GalNAc and UDP-Glc [91]. More recently, three different UDP-sugars (UDP-GlcA, UDP-GlcNAc and UDP-GalNAc) were shown to be transported into the Golgi lumen by the SLC35A5 protein [92].

Multi-specific NSTs were first thought to carry monosaccharides activated by only one type of diphosphonucleotide (i.e., either UDP-sugars or GDP-sugars). However, an NST specific for UDP-Xyl, UDP-GlcNAc and GDP-Fuc was identified in the fruit fly [93]. This is the first and, so far, only report on an NST translocating both UDP- and GDP-sugars.

Initial reports suggested that the transport of individual NSs by multi-specific transporters is a competitive process [64]. According to this view, different NSs would be transferred via the same active site. The *C. elegans* NST encoded by the *CO3H5.2* gene is specific for UDP-GlcNAc and UDP-GalNAc [66]. Kinetic studies on a protein over-expressed in *S. cerevisiae* have shown, however, that both NSs are transferred to the Golgi lumen independently. The deletion mutant of this NST, lacking 16 amino acid residues located within the loop between the second and third TMDs, lost the ability to transport UDP-GalNAc but retained the ability to carry UDP-GlcNAc. The authors concluded that different portions of this NST were involved in transfer of distinct NSs. A similar phenomenon of independent transport of two different NSs was observed for the UDP-GlcNAc/UDP-Gal-specific SRF-3 protein from *C. elegans* [94].

### 3.4. Oligomerization of NSTs

According to numerous studies, NSTs were shown to form dimers or higher oligomers. In vitro dimer formation was shown for the rat UDP-GalNAc [73] and GDP-Fuc [74] transporters and for the yeast GDP-Man transporter [78,84,95]. The GDP-Man transporter from *L. donovani* was shown to form hexamers in vitro [88]. Oligomeric structures were also formed in vitro by the canine UDP-Gal transporter [96]. Moreover, SLC35A3 [97], SLC35A5 [92] and SLC35A1 [98] proteins were shown to dimerize in living cells.

It is not entirely clear which polypeptide fragments participate in dimerization of NSTs. The amino acid sequences of the mammalian UDP-Gal (SLC35A2) and CMP-Sia (SLC35A1) transporters and the yeast UDP-GlcNAc transporter contain a leucine zipper motif [99]. This motif was shown to mediate dimerization of certain proteins, but not all NSTs that tend to dimerize contain this sequence [88,95]. Moreover, the mouse CMP-Sia transporter lacking a leucine zipper motif was shown to be fully functional [77]. In the case of the yeast GDP-Man transporter (Vrg4) the C-terminal TMD was shown to be indispensable for dimerization [95].

Environmental factors were also shown to play a role in the Vrg4 dimerization. Specifically, dimerization of Vrg4 was found to be mediated by lipids as their presence was revealed at the dimer interface [78]. The effect of point mutations in the SLC35A1 gene on the dimerization capacity of the CMP-Sia transporter was also examined [98]. This study revealed that disease-causing mutations, Q110H and E196K, tend to impair/prevent dimerization of this NST.

### 3.5. Antiport Mechanism of NS Translocation 

To explain the mechanism of NS transport, a model of electroneutral antiport was proposed, during which the NS is transferred to the ER and/or Golgi lumen and the corresponding nucleotide monophosphate (NMP) is transferred to the cytoplasm [51,100,101,102]. It was proposed that the nucleotide diphosphates (NDPs) formed after the transfer of the sugar residues onto the acceptors are degraded by organellar nucleotide diphosphatases (NDPases) to NMP and inorganic phosphate [48,101,103,104]. This reaction is important not only because it generates the compounds to be antiported but also due to the fact that NDPs are inhibitors of glycosyltransferases [105]. The transport constant K_m_ for the majority of NSTs is in the range of 1–10 µM. The antiport model was recently supported by the crystal structures of yeast GDP-Man [78], mouse CMP-Sia [79] and *Zea mays* CMP-Sia [106] transporters.

It was proposed that NS transport results in the accumulation of NSs in the Golgi lumen [45,48,51]. However, the concept of accumulation appears to be challenged by the 1:1 antiport model as the former does not explain how transport into the lumen of the Golgi vesicles can be sustained without stoichiometric production of the corresponding NMP.

Several reports suggests that exchange for the corresponding NMP is not an absolute prerequisite for NS transport to occur. Deletion of a gene encoding the yeast guanosine diphosphatase resulted in a reduction in the amount of mannosylated N-glycans [107]. However, a different study showed that the biosynthesis of mannosylated N-glycans in yeast lacking functional guanosine diphosphatase was not significantly inhibited [108].

The results of some studies suggest that the transport of selected NSs is facilitated by an exchange of the corresponding NDP [51,89] or, alternatively, another type of NS [51,53,54,89]. The data obtained by Bossuyt and Blanckaert suggest the possibility that NSs can be transported in both directions. Specifically, pre-incubation of rat microsomes with UDP-GlcNAc stimulated UDP-GlcA import [54]. Based on these results the presence of a conjugated system of two transporters in the ER membrane was proposed, one of which would import UDP-GlcA with simultaneous export of UDP-GlcNAc and the other would import UDP-GlcNAc with concomitant export of UMP molecule formed upon incorporation of GlcA into glycoconjugates followed by UDP breakdown [109]. This phenomenon would be restricted to the ER as it was not observed for the Golgi apparatus [110].

### 3.6. Pathologies Related with Defective NSTs

Defective function of several NSTs leads to some disorders in humans. Several mutations in the gene encoding the GDP-Fuc transporter result in a disease termed CDGIIc (congenital disorder of glycosylation type IIc) or LADII [50,111]. This disease is manifested by decreased fucosylation of many glycans, including blood group antigens and selectin ligands [112,113,114]. The level of α-1,6-fucosylation of N-glycans (the so-called core fucosylation) is particularly reduced in CDGIIc/LADII patients [115]. The affected individuals suffer from dysfunctions of the immune system and exhibit developmental delay [116]. Surprisingly, some of the symptoms become alleviated in response to oral administration of Fuc, which is one of the GDP-Fuc precursors [113,117,118,119]. This effect suggests either the partial activity of the mutant transporters or the existence of alternative mechanisms of GDP-Fuc transport into the Golgi apparatus.

In 2005, a mutation in the gene encoding the CMP-Sia transporter was identified and the resulting disorder was termed CDGIIf [41,120]. Nowadays, the corresponding conditions are classified as SLC35A1-CDG because the CMP-Sia transporter is encoded by the *SLC35A1* gene. At the molecular level, the lack of sialyl Lewis X antigen, a selectin ligand, on the surface of the patient-derived multinucleated granulocytes was demonstrated. Subsequently, more cases of SLC35A1-CDG were characterized [121,122,123]. The affected individuals displayed neurological symptoms such as intellectual disability, hypotonia, ataxia and seizures as well as macrothrombocytopenia.

In 2006, a disease related with a point mutation in the *SLC35A3* gene, encoding the UDP-GlcNAc transporter, was characterized in cattle [124]. The disease was termed CVM (Complex Vertebral Malformation) as the main symptoms included severe spine and rib anomalies. In 2017, a compound heterozygous mutation in the human *SLC35A3* gene was linked to severe epileptic encephalopathy with skeletal abnormalities [125]. Mutations in this gene have also been linked to autism [126].

The human ER-resident UGTrel7/SLC35D1 transporter was shown to be specific for UDP-GlcA and UDP-GalNAc [60]. In mice, the knockout of the corresponding gene leads to a lethal form of skeletal dysplasia [127]. In the affected animals the presence of truncated chains of chondroitin sulphate in proteoglycans was observed. In humans, mutations in the *SLC35D1* gene cause a severe form of skeletal development abnormality, known as Schneckenbecken dysplasia [127,128].

Mutations in the *SLC35A2* gene encoding the UDP-Gal transporter have also been linked to several pathologies including SLC35A2-CDG e.g., [129,130,131,132,133,134]. The affected individuals display neurological symptoms such as epilepsy, encephalopathy and hypotonia, dysfunctions of the liver, spleen, and kidneys as well as skeletal abnormalities.

## 4. Studies of Nucleotide Sugar Transporters in Mammalian Cells

Nearly 50 years of research on translocation of NSs has provided multiple pieces of information that need to be consolidated in order to fully understand the process of glycosylation in mammals. The current general picture proposes several different NSTs supplying major glycosylation substrates in mammals. If valid and complete, the knowledge gathered using multiple different systems and approaches should allow to make hypotheses and predictions in vivo.

The main research efforts of our laboratory are dedicated to understand the process of glycosylation in mammalian cells with an emphasis on the role of the solute transporters from the SLC35 subfamily. We also aim to understand molecular bases of the monosugar supplementation therapies that are often successfully applied to slow down progression of certain CDG types.

Our leading strategy is to use the CRISPR-Cas9-assisted gene inactivation system, to generate mammalian knockout cell lines deficient in individual *SLC35* genes. In some cases, double knockouts, such as *SLC35A2*/*SLC35A3* or *SLC35C1*/*SLC35C2*, have also been generated. Subsequently, the glycophenotypic effects of the *SLC35* knockouts have been studied using a variety of techniques ranging from lectin staining, through transport of radiolabelled NSs up to mass spectrometry analysis of cellular (surface and secreted) glycans.

In addition, fluorescence- and luminescence-based techniques including in situ *PLA*, FLIM-FRET, BIFC-based FRET and NanoBiT as well as conventional co-immunoprecipitation are applied to study formation of complexes. Using these strategies, we investigated several NSTs in a set of mammalian cell lines including HepG2, HEK293T, PC-3, MDCK-RCA^r^ and COS-7.

Below we describe our most important observations regarding transport of selected NSs, including UDP-Gal, UDP-GlcNAc, CMP-Sia, GDP-Fuc and UDP-Xyl, and summarize our findings with respect to the complexes formed by several selected NSTs. The results are also collectively presented in Figure 1, which provides a graphical summary of the postulated NS transport routes in the mammalian Golgi. Aside from our results, we also provide a chapter summarizing the complexity and some inconsistencies concerning the generally accepted antiport mechanism of NS translocation.

### 4.1. UDP-Galactose Supply

To date, the only UDP-Gal transporter identified in mammalian cells is SLC35A2 (UGT) [32,33,96,135]. A detailed functional characterization of SLC35A2 was possible after respective mutant cell lines, such as MDCK-RCA^r^ [28] or CHO-Lec8 [17,135], had been generated. Mutations identified in these cells abolish production of the functional protein, giving rise to glycoconjugates deficient in Gal and Sia.

The expression of the *SLC35A2* gene results in two splice variants: Golgi-resident UGT1 and UGT2, which localizes both to the ER and Golgi [136]. Our team showed that both UGT1 and UGT2 were able to restore wild-type glycophenotype in MDCK-RCA^r^ and CHO-Lec8 cells with impaired UDP-Gal transport [137]. Using a CRISPR/Cas9 approach, we also generated human cell lines (HepG2 and HEK293T) deficient in SLC35A2 activity [138]. Both studies revealed that a subset of N-glycans synthesized by the SLC35A2-deficient cells contains Gal. This suggests that there may be a different transport route for UDP-Gal than via SLC35A2.

Interestingly, as judged by lectin staining and MALDI-TOF analysis of the cellular N-glycan structures, a galactosylation defect in the MDCK-RCA^r^ and CHO-Lec8 cells could be partially restored by the over-expression of the canine SLC35A3 [139]. This implies that SLC35A2 and SLC35A3 may be functionally connected and that SLC35A3 may play some role in the transport of UDP-Gal.

We have also produced a human chimeric protein composed of the amino acids 1–224 of the SLC35A2 C-terminally fused to the amino acids 198–325 of the SLC35A3 [140]. Such a construct was over-expressed in the MDCK-RCA^r^ and CHO-Lec8 cells and, to our surprise, it was not only functional, but also was targeted to the Golgi apparatus and appeared to fully restore the wild-type galactosylation phenotype of both cellular N- and O-glycans.

To map the minimal fragment of the SLC35A2 protein sufficient to ensure the wild-type glycosylation phenotype, four additional SLC35A2/SLC35A3 chimeric proteins were investigated [141]. The constructs were prepared such that the contribution of the SLC35A2 transporter was successively reduced up to only 35 N-terminal amino acids, while the contribution of the SLC35A3 was gradually increased. Over-expression of these chimeras in the MDCK-RCA^r^ and CHO-Lec8 showed that already the construct with the smallest contribution of the SLC35A2 was sufficient to correct the galactosylation phenotype. Importantly, analogous chimera composed of amino acids 1–35 of the SLC35A2 and amino acids 12–337 of another homologous SLC35A family member, SLC35A1, failed to restore the wild-type phenotype. This suggests that the fragment of the SLC35A3 may play a role in the translocation of UDP-Gal.

Further evidence that SLC35A3 may be involved in the transport of UDP-Gal came from gene silencing experiments with siRNA against *SLC35A3* sequence performed in the CHO cells [142]. Unexpectedly, transport of the radiolabelled UDP-Gal into the Golgi vesicles was more severely affected than the uptake of UDP-GlcNAc, which is considered the main substrate for SLC35A3. However, this outcome could not be reproduced in the *SLC35A3* knockout CHO cells. Here, a decrease of transport of UDP-Gal into Golgi vesicles upon disruption of the *SLC35A3* gene was not observed [138].

The effect of the *SLC35A2/SLC35A3* double knockout on the galactosylation phenotype is even more complicated. Mass spectrometry (MS) analysis of the N-glycans from the double knockout HEK293T cells confirmed the absence of Gal residues in the cellular glycans, while the N-glycans released from the secreted reporter glycoprotein, SEAP, were galactosylated [138]. The aforementioned observations could be rationalized by an alternative UDP-Gal delivery system for galactosylation of the secreted acceptors. However, it cannot be excluded that the secreted glycoproteins obtained Gal residues after leaving Golgi, e.g., in the secretory vesicles or even in the medium. Alternatively, the presence of Gal residues in SEAP-derived N-glycans may be cell type-specific as no galactosylated structures were observed for the *SLC35A2/SLC35A3* double knockout in the HepG2 and CHO cells.

Taken together, further studies are necessary to fully understand the UDP-Gal delivery into the Golgi lumen. Undoubtedly, it appears to be a complex process that is possibly mediated by more than one transporter protein with an emphasis on the potential involvement of the SLC35A3 kin.

### 4.2. UDP-N-Acetylglucosamine Supply

The UDP-GlcNAc is an important glycosylation substrate because, similarly as for UDP-Gal, GlcNAc is also found in all major classes of glycoconjugates including N-glycoproteins, O-glycoproteins, proteoglycans and glycolipids. Among the mammalian proteins proposed to translocate UDP-GlcNAc to the Golgi lumen are: (i) SLC35A3 (discovered earliest), (ii) SLC35D2, and (iii) SLC35B4.

Mammalian SLC35A3 was discovered by Guillen et al. in 1998 by complementation of the *K. lactis* mutant unable to translocate UDP-GlcNAc across the Golgi membranes with the canine *SLC35A*3 gene [34]. The human *SLC35A3* gene was identified based on nucleotide sequence similarity to the human *SLC35A2* gene and was demonstrated to localize to the Golgi apparatus in the CHO cells [59]. The transporter was shown to be specific towards UDP-GlcNAc and not to UDP-Gal and CMP-Sia in the *S. cerevisiae* heterologous system.

In the Golgi lumen, GlcNAc is incorporated into N-glycan antennae by mannoside N-acetylglucosaminyltransferases. The Mgat1 and Mgat2 form mono- and biantennary N-glycans, while Mgat4 and Mgat5 are responsible for further branching [143]. In order to test whether SLC35A3 delivers UDP-GlcNAc for glycan biosynthesis, we silenced the *SLC35A3* gene using siRNA technology in the CHO, HeLa and MDCK cell lines [142]. As GlcNAc is an important component of the complex-type N-glycans, we anticipated that the *SLC35A3* knockdown would result in a severe impairment of these structures. To our surprise, only the tri- and tetra-antennary species were depleted, while the bi-antennary ones were unperturbed. Hence, we concluded that the SLC35A3 transporter may selectively supply Mgat4 and Mgat5 transferases, while Mgat1 and Mgat2 are supplied with UDP-GlcNAc by another transporter.

The effect of *SLC35A3* knockdown was also investigated in relation to the synthesis of keratan sulphate (KS) proteoglycans [142]. This type of GAG is composed of alternating units of Gal and GlcNAc. Relative to other cell lines, MDCK produces particularly large quantities of KS proteoglycans which makes it a good model to investigate these glycoconjugates. As anticipated, the amount of KS proteoglycans was significantly depleted in the MDCK cells with an *SLC35A3* knockdown. On the other hand, heparan sulphate (HS), which is composed of alternating units of GlcNAc and GlcA, was not affected. This difference may be due to the selective supply of UDP-GlcNAc by SLC35A3 to KS, and not to HS, synthesis.

Although our results obtained using siRNA technology provided some insights into the role of SLC35A3 in the synthesis of cellular N- and proteoglycans, some of the observed effects might have been underestimated due to the incomplete gene silencing. Therefore, as a follow up, we knocked out the *SLC35A3* gene using the CRISPR/Cas9 approach in mammalian cell lines, i.e., HepG2, HEK293T and CHO. Similar to the previous study [142], we observed that *SLC35A3* knockout resulted in a decrease in the amount of tri- and tetra-antennary N-glycans regardless of the analysed cell line, while bi-antennary species remained unaffected. The depletion of the multibranched N-glycans in *SLC35A3* knockouts was potentiated by the lack of SLC35A2.

In the same study, we also investigated N-glycans decorating secreted SEAP reporter glycoprotein. Here, a very mild to no effect on the level of multibranched structures was observed for the single *SLC35A3* as well as for the double *SLC35A2/SLC35A3* knockouts.

In addition to N-glycans and some proteoglycans, GlcNAc is also present in the structures of some O-linked glycans. As judged from the MALDI-TOF spectra, essentially all GlcNAc-containing O-glycan structures produced by the wild-type cells were also present in the samples derived from the *SLC35A3* knockouts. Taken together, these results may suggest redundancy of SLC35A3 in the supply of UDP-GlcNAc for N- and O-glycan biosynthesis.

The *SLC35A3* knockouts were also tested for the ability to translocate radiolabelled UDP-GlcNAc into Golgi vesicles in HEK293T and CHO cell lines but the results were inconclusive, i.e., a decrease of the rate of transport observed for the HEK293T cells could not be reproduced in the CHO cells [138].

The UDP-GlcNAc is also a substrate for O-GlcNAcylation, a post-translational modification that takes place in the cytoplasm and nucleus [3], which was recently shown to also affect SLC35A3 [144]. The knockdown of the O-GlcNAc transferase (OGT), the only known enzyme responsible for O-GlcNAcylation, decreased synthesis of tri- and tetra-antennary N-glycans, which is a similar effect to the *SLC35A3* knockout. These results by Song et al., obtained for HeLa cells [144], are in line with our results for CHO cell line [138,142]. The study also showed that SLC35A3 transporter’s activity can be regulated by the cytoplasmic O-GlcNAc modification.

Apart from SLC35A3, UDP-GlcNAc appears to be translocated to the Golgi by SLC35D2 (also known as HRFC1) [61]. The gene was identified by homology to the *D. melanogaster frc* and *C. elegans sqv-7*. The SLC35D2 was shown to localize to the Golgi of the HCT116 cells and to be specific for UDP-GlcNAc and UDP-Glc in the *S. cerevisiae* heterologous system. Over-production of SLC35D2 in the HCT116 cells increased the amount of GlcNAc-rich HS, which suggests that SLC35D2 may be involved in supply of UDP-GlcNAc for HS synthesis.

The UDP-GlcNAc was also shown to be transported by the human SLC35B4 over-produced in yeast [63], however, in the native context the protein appears to localize to the ER (described in Section 4.5) and it seems to be redundant for UDP-GlcNAc delivery for biosynthesis of glycoconjugates [145]. In summary, supply of UDP-GlcNAc for glycosylation remains poorly understood but surely relies on more than one transporter protein.

### 4.3. CMP-Sialic Acid Supply

The only so far identified mammalian CMP-Sia transporter (CST) is SLC35A1 [36,146]. To date, mouse SLC35A1 is the only mammalian NST whose three-dimensional structure has been determined [79]. The amino acid sequence similarity to the human variant is about 91%. Structural data demonstrated that SLC35A1 contains 10 transmembrane helices with both N- and C-termini facing the cytosol. Both TMD5 and TMD10 appear to be involved in dimer formation, while the other domains seem to be involved in the formation of transport bundles [106].

The first studies on SLC35A1 began with the isolation of two mutants from a CHO cell line based on their resistance to WGA. CHO-Lec2 [15] and clone 1013 [18] were isolated independently and both showed a dramatic reduction in sialylation caused by mutations in the *SLC35A1* gene [147]. The over-expression of human SLC35A1 in CHO-Lec2 cells enabled restoration of the wild-type phenotype and the microsomal vesicles isolated from these cells exhibited high CMP-Sia transport activity [36]. Moreover, a murine SLC35A1 expressed in *S. cerevisiae* gave them the ability to transport CMP-Sia into Golgi vesicles [56]. In addition, the CHO MAR-11 cell line was developed. These cells contain a point mutation in the *SLC35A1* gene, resulting in a premature stop codon [148] and their phenotype is characterized by a yet lower surface sialic acid than CHO-Lec2 cells.

Recently, we knocked out the *SLC35A1* gene in the human cell line HEK293T using CRISPR-Cas9 strategy [98]. Although the level of sialylated structures was reduced, sialylated N-glycans were still detectable. Moreover, a sialylated glycosphingolipid species (GM3) was synthesized by the SLC35A1-deficient cells. Altogether, these results suggest the existence of an SLC35A1-independent Golgi CMP-Sia uptake route. 

### 4.4. GDP-Fucose Supply

Fucose is a terminal residue found in N- and O-glycans. So far, two GDP-Fuc transporters have been identified in humans, i.e., SLC35C1 and SLC35C2.

In 1999 Lübke et al. associated a genetic disorder, LADII, resulting in a general decrease in fucosylation, with an ineffective translocation of GDP-Fuc to the Golgi apparatus [50]. Based on these findings, in 2001, two independent groups identified genes encoding human GDP-Fuc transporter, SLC35C1 [37,111]. Lübke et al. characterized the human transporter protein by complementation of fibroblasts from a LADII patient with cDNA library constructed from human liver [111].

Independently, Lühn et al. cloned *C. elegans* genes into the cells of a LADII patient and observed restoration of fucosylation [37]. Then, by sequence comparison of the genes encoding for putative GDP-Fuc transporters, they identified human gene, *SLC35C1*. Moreover, they showed that SLC35C1 localized to the Golgi apparatus. It was also shown that upon oral administration of Fuc to the LADII patients, fucosylation of macromolecules was restored [114,117,119].

The first attempt to resolve the phenomenon of the success of the oral Fuc therapy was done by Hellbusch et al. who studied the effect of Fuc supplementation in mice deficient in SLC35C1 protein [149]. They showed that the mechanism of Fuc treatment leading to the improvement in fucosylation is independent of SLC35C1, which suggests an existence of an alternative GDP-Fuc translocation path to the Golgi.

Studies of the mechanism of Fuc supplementation were also attempted by our laboratory [150]. We generated *SLC35C1* knockouts in two human cell lines, HEK293T and HepG2. The effect of the GDP-Fuc transporter deficiency was quantified using our customized HPLC method in which all multibranched complex-type N-glycans are first enzymatically reduced to a simple biantennary (GlcNAc)_2_(Man)_3_ structure (either fucosylated or non-fucosylated) and then separated on a column. The percentage of fucosylation is calculated as a ratio of the signal from the fucosylated glycan divided by the sum of the signals from both fucosylated and non-fucosylated species.

Using our method, we quantified the percentage of fucosylation of the cellular complex-type N-glycans in the wild-type HEK293T and HepG2 cells to be ~80%. In the *SLC35C1* knockouts the fucosylation was reduced to ~8% and ~15%, respectively.

In this approach we could not, however, exclude that the fucosylated structures found in the *SLC35C1* knockouts came from fucosylated glycoconjugates contained in the serum. Therefore, we employed a close to zero-background system in which the analysis is performed on the purified N-glycans decorating His-tagged SEAP reporter over-produced in the cells. Here, thanks to the washing of the Ni-NTA-bound SEAP, the possibility of the presence of serum-derived glycoconjugates is minimized. Using this system, we demonstrated that the remaining ~8–15% of fucosylation is not due to the contamination of the sample by serum glycoconjugates but indeed a result of fucosylation occurring in the *SLC35C1* knockout cells.

In the same study we also analysed secreted O-glycans using MALDI-TOF mass spectrometry [150]. Our spectra show that in the *SLC35C1* knockout cells fucosylation was significantly reduced, but not completely abolished.

As Fuc feeding is a basis of LADII therapy, the analysis of fucosylation phenotype of the wild-type and *SLC35C1* knockout was repeated for the cells supplemented with exogenous Fuc. First, we optimized supplementation conditions, i.e., time of treatment and Fuc concentration in culture media. We observed that the addition of 5 mM Fuc to culture media nearly completely restored fucosylation of both N- and O-glycans in the *SLC35C1* knockouts [150].

It was speculated that Fuc treatment causes an increase in GDP-Fuc concentration in cytosol of patients’ cells, thereby forcing defective SLC35C1 variants to translocate more GDP-Fuc to the Golgi lumen [151]. We showed that the basal level of GDP-Fuc in the wild-type and in the *SLC35C1* knockout is the same (~5–13 μM) and increases ~40–50 fold upon feeding with 5 mM Fuc to ~150–200 μM [150]. Millimolar concentrations of exogenous Fuc were required to observe restoration of the wild-type fucosylation in the *SLC35C1* knockout cells.

In summary, our results demonstrate that fucosylation of N- and O-glycans can be restored by supplementation with millimolar concentrations of Fuc not only in the cells with pathogenic variants of the *SLC35C1* gene, but also in the cells completely lacking SLC35C1.

In mammalian cells, GDP-Fuc is synthesized via two biosynthetic pathways, i.e., de novo and salvage. The de novo pathway utilizes Man as a substrate, while the salvage pathway uses Fuc recovered from lysosomal degradation of glycoconjugates or Fuc obtained from the environment [152]. Having determined that high Fuc concentration could restore fucosylation in *SLC35C1* knockout cell lines, we checked whether an addition of high (5 mM) amounts of Man would also increase intracellular GDP-Fuc concentration. The outcome of this experiment could either support or disprove the concept of oral administration of Man as an alternative treatment for LADII patients. We observed that 5 mM Man added to the culture medium caused no changes in the intracellular GDP-Fuc concentration neither in *SLC35C1* knockout nor in the wild-type cells.

Aside from phenotypic investigation of the *SLC35C1* knockout with and without Fuc supplementation, we have also followed metabolic fate of the exogenous monosugars (Fuc and Man). Radiolabelled compounds allowed detection of the residual fucosylation of N-glycans synthesized by the *SLC35C1* knockout cells. By feeding the cells with nanomolar concentrations of either of the radioactive monosugars, [^3^H]Fuc or [^3^H]Man, and comparing radioactivity of the fucosylated N-glycans, we found that the SLC35C1-deficient cells preferentially used GDP-Fuc produced in the salvage pathway as compared with the de novo pathway. This was the first indication that an NST could discriminate between the substrates coming from different sources.

Taken together, we proposed an existence of three different GDP-Fuc transport systems in the mammalian Golgi membrane. The first one, SLC35C1-dependent, mainly utilizes the NS pool derived from the de novo pathway. The other two, independent form SLC35C1, mainly use GDP-Fuc synthesized in the salvage pathway. Among the latter two, one is able to work at basal levels of intracellular GDP-Fuc, whereas the other requires much higher substrate concentrations that can only be achieved by feeding the cells with millimolar Fuc [150]. The identity of those alternative GDP-Fuc transport systems remains yet to be discovered.

The GDP-Fuc was also reported to translocate to the Golgi via another transporter, SLC35C2 [93,153], which has about 22–23% identity and 37–38% similarity to SLC35C1 and mainly localizes to the Golgi apparatus [153].

As mentioned above, cells completely deficient in SLC35C1 retained small but non-negligible ability to incorporate Fuc into their N- and O-glycans. To investigate whether the remaining fucosylation may be supported by SLC35C2, our group generated double *SLC35C1/SLC35C2* knockout in the HEK293T cells [150]. Even in the absence of both SLC35C1 and SLC35C2 the fucosylated glycans were still detectable to the same level as for the SLC35C1 single knockout. This may indicate that either SLC35C2 is not the backup GDP-Fuc supplier for fucosylation of the N- and O-glycans in the HEK293T cells or its deficiency is compensated by another NST(s) of yet unknown identity.

### 4.5. UDP-Xylose Supply

In mammals, Xyl is found in a tetra-saccharide linker that links GAG chains with core proteins in proteoglycans (-GlcA-Gal-Gal-Xyl-Ser/Thr). To date, the only known mammalian transporter of UDP-Xyl is SLC35B4. This protein comes in two splice variants: a longer version encoding a protein of 331 amino acids and a shorter version encoding a protein of 231 amino acids.

In 2005, Ashikov et al. showed that vesicles from *S. cerevisiae* expressing the longer isoform of human SLC35B4 displayed a specific uptake of UDP-Xyl and UDP-GlcNAc [63]. However, research on a homologue from *D. melanogaster* showed ability to also transport GDP-Fuc [93]. Moreover, microsomes derived from Chinese hamster V79 cells over-expressing either of the splice variants were able to also transport UDP-GlcA, but only after preloading of the microsomes with UDP-GlcNAc [154].

The subcellular localization of SLC35B4 has originally been assigned to the Golgi apparatus by over-expression of the FLAG-tagged longer isoform of human transporter in CHO cells, which suggests that this protein might be involved in transport of UDP-Xyl and UDP-GlcA that serve as substrates for proteoglycan synthesis [63]. However, in 2011, our team over-expressed FLAG-tagged SLC35B4 protein in MDCK and MDCK-RCA^r^ cell lines and showed that both splice variants co-localized exclusively with calnexin, an ER marker [155].

In another study, we showed that lysine 329 within a C-terminal dilysine motif KDSKKN is crucial for the ER localization of the human SLC35B4 [145]. Moreover, C-terminal tagging resulted in Golgi localization of the over-expressed SLC35B4. These results indicate that NST tagging may lead to abnormal localization and should be used with caution. In the same study, we generated SLC35B4 knockout in the HepG2 cell line using the CRISPR-Cas9 approach. The SLC35B4-deficient cells were of the wild-type phenotype with respect to the glycoprotein and proteoglycan structures. These observations suggest that, despite the specificity towards UDP-Xyl and UDP-GlcNAc determined in yeast heterologous system [63], SLC35B4 does not seem to provide these substrates to the mammalian Golgi. First, as it is localized to the ER and second as it is redundant for biosynthesis of mammalian glycoconjugates containing Xyl and GlcNAc residues.

### 4.6. Mechanism of Transport of Nucleotide Sugars

Nucleotide sugar transporters are believed to act as antiporters, exchanging cytosolic NS for a corresponding lumenal NMP [87]. This was first postulated by Capasso and Hirschberg in 1984 who treated Golgi fractions isolated from rat liver with GDP-Fuc labelled with tritium in the guanosine ring [156]. When the resulting vesicles were supplemented with GDP-Fuc labelled with ^14^C in the monosaccharide moiety, a decrease of tritium signal within the vesicles was observed. The authors also showed that it was caused by the exit of [^3^H]GMP from the Golgi vesicles.

However, experiments using *S. cerevisiae* with a null mutation in the gene encoding for GDPase, an enzyme required for protein and sphingolipid mannosylation, showed that reduced level of lumenal GMP caused ~5-fold reduction but did not completely abolish GDP-Man uptake into the Golgi fractions [157].

The mechanism of antiport was also studied for other NSs e.g., UDP-Gal, UDP-GlcA and UDP-Xyl [72]. Here, NS transport activities were reconstituted into artificial liposomes composed of PC or Golgi lipids. In the absence of UMP, the transport of the UDP-sugars into the lumen occurred only until solute equilibration but no accumulation was observed. When the vesicles were pre-loaded with UMP, translocation of the UDP-sugars was ~2–3-fold more efficient than without the pre-loading. This may indicate that the translocation of the UDP-sugar across the membrane can occur in the absence of UMP, however, the presence of some residual NMP and/or NDPase activity inside the vesicles cannot be excluded.

This problem seems to be resolved by Nji et al. who investigated CMP-Sia uptake into the proteoliposomes composed of brain lipids and purified human SLC35A1 [106]. Here, the presence of residual NMP within the proteoliposomes was excluded and the system proved low to zero background. In the presented setup, minute concentrations of CMP-Sia were able to translocate into the vesicles in the absence of the transporter. The vesicles with the reconstituted SLC35A1 translocated CMP-Sia and this transport was highly stimulated by the pre-loading of the vesicles with CMP. Based on these results, a model was proposed, in which SLC35A1 can operate as both passive and active antiporter.

It is worth mentioning experiments conducted by Waldman and Rudnick in 1990 [51]. According to them, previous studies [156] might have overestimated the exchange rate of the internal [^3^H]GDP-Fuc for the external GDP-[^14^C]Fuc. This hypothesis was based on their observation that ~15% of the [^3^H]UDP-GlcNAc internalized to the Golgi-enriched vesicles hydrolysed to [^3^H]GlcNAc-1-phosphate [51]. Interestingly, transport of [^3^H]UDP-GlcNAc could also occur in the opposite direction, i.e., outside of the vesicles and was stimulated by not only UDP-GlcNAc but also by UMP, UDP, UDP-Gal, but not by AMP.

The antiport mechanism was also studied for UDP-GlcNAc and UDP-Gal in the semi-permeabilized mouse thymocytes [158]. In this study, addition of the external 1 mM UMP/ UDP caused efflux of intravesicular UDP-[^14^C]GlcNAc. The fact that the effect of UDP was about twice as strong as the effect of UMP may suggest that, in addition to the original model of equimolar NS:NMP exchange, NDP may also serve as the antiported molecule. Indeed, intravesicular [^3^H]UDP stimulated uptake of both UDP-GlcNAc and UDP-Gal.

These reports strongly suggest that the initially simple and elegant view of equimolar exchange of cytoplasmic NS for the lumenal NMP must be extended by several other potential transport mechanisms. Among the most important alternative scenarios that must be considered are: (i) NS exchange for NDP, (ii) NS exchange for another NS, (iii) uniport of NS in the absence of the antiported molecule (passive antiport).

### 4.7. Homo- and Heterologous Complexes of NSTs

There are many reports in the literature about the ability of the transporters to form homo- and heterodimers (partially described in Section 3.4) including several discovered by our laboratory (described below). 

In our group complex formation between NSTs was studied using multiple experimental approaches. Using co-immunoprecipitation and FLIM-FRET, we showed the existence of an SLC35A2/SLC35A3 complex in the MDCK-RCA^r^ cells [97]. Later on, this interaction was confirmed in the HepG2 cells using in situ PLA [159] and in the HEK293T cells using NanoBiT [160]. Such consistent colocalization of these two transporters demonstrated in multiple cell lines using different experimental techniques may suggest their functional connection.

Hetero-oligomers are also formed by an orphan NST, SLC35A4. Sosicka et al. showed that SLC35A4 oligomerizes with its kin, SLC35A5 [159]. Moreover, in SLC35A4 knockout HepG2 cells, the localization of the SLC35A2/SLC35A3 heteromer was perturbed. In another study a ternary complex formed by SLC35A2, SLC35A3 and SLC35A4 was reported in COS-7 cells using BIFC-based FRET [161].

Aside from formation of homo- and hetero-oligomeric complexes with their kins, NSTs were also found to associate with glycosyltransferases (GTs). The SLC35A2 was shown to associate with the ceramide galactosyltransferase (UGT8) [86]. Using FLIM-FRET and in situ PLA our group demonstrated formation of binary complexes between either SLC35A2 (both spice variants) or SLC35A3 and Mgats 1–5 in three different mammalian cell lines including PC-3, MDCK-RCA^r^ and HEK293T [162]. Furthermore, several ternary complexes composed of a single NST (SLC35A2, SLC35A3 or SLC35A4) and two different Mgats or two NSTs and a single Mgat were detected using BIFC-based FRET in COS-7 cells [161]. In addition to Mgats, SLC35A2 was also shown to form binary complexes with galactosyltransferases B4GalT1 [163] and B4GalT4 [164] in HEK293T cells using NanoBiT. Such a rich representation of hetero-oligomeric complexes formed between NSTs and GTs, that are directly involved in the attachment of monosugar residues to the nascent glycans, may indicate a potential functional coupling of NS transport and glycan biosynthesis.

Moreover, in 2021, using pull-downs coupled with mass spectrometry we identified a set of proteins that may interact with three SLC35A subfamily members [165]. In this experiment over-expressed SLC35A2-A4 was used as bait to co-immunoprecipitate potential interaction partners from HepG2 cell lysates. For each of the NST ~20–30 interacting proteins were identified including ATPases, ion channels/transporters, lipid metabolism/membrane insertion/translocation enzymes as well as chaperons and protein transport receptors. For a small set of selected candidates, the interactions were verified using NanoBiT system. The SLC35A2, SLC35A3 and SLC35A4 were showed to interact with e.g., Golgi pH regulator B (GPR89B) and ATPase 2 (ATP2A2). These observations suggest that NSTs may associate with a wider range of functionally distinct membrane proteins potentially involved in regulation of glycan biosynthesis.

For the case of NSTs homo-oligomerization, the research of our group was focused mainly on SLC35A2, SLC35A3 and SLC35A1. Formation of homo-oligomers by over-expressed canine SLC35A2 in CHO cells was confirmed using co-immunoprecipitation [96]. Similar observations were made for the human homologue in HEK293T cell using the NanoBiT system [163]. For SLC35A3, homo-oligomers were observed in MDCK-RCA^r^ cells using FLIM-FRET [97]. In the case of SLC35A1, homo-oligomerization was shown in HEK293T using NanoBiT [98].

In the latter study, we also investigated the effect of selected point mutations known to cause SLC35A1-CDG. Two disease-causing SLC35A1 single amino acid mutants, i.e., Q101H in TMD3 and E156K in TMD6 were unable to produce luminescence signal in the NanoBiT experiment. Our observations show that the loss of function of those mutants was accompanied by the inability to dimerize in the Golgi membrane of living cells.

The role of interactions between NSTs and other proteins is not fully understood. There are more and more reports about such interactions, but very little is known about the biological significance of these phenomena. We believe that NSTs might be hub proteins that are in the centres of glycosylation-related interaction networks also containing GTs and some regulatory proteins. Such assemblies would be functionally complete and self-sufficient units in the Golgi membranes, in which the NS supply could be spatially and temporarily coupled to glycosylation to achieve higher efficiency and fidelity of glycan synthesis. 

### 4.8. Miscellaneous Controversial Data

Apart from the aforementioned uncertainties associated with the delivery of CMP-Sia, UDP-Gal, UDP-GlcNAc, GDP-Fuc and UDP-Xyl to the Golgi we would like to point at some other missing puzzles of the NS transport.

To date, the delivery of UDP-GalNAc for the biosynthesis of mucin-type O-glycans (O-GalNAc glycans) has not been assigned to any specific member of the SLC35 family. The UDP-GalNAc was shown to be transported by the UGTrel7/SLC35D1 protein [60]. Nevertheless, this NST was shown to localize to the ER, whereas the initiation of mucin-type O-glycan biosynthesis occurs in the Golgi [6]. Hence, it appears unlikely that SLC35D1 could fulfil this role unless one assumes that NSs traffic between the organelles via vesicular transport.

In another study, UDP-GalNAc was proposed to translocate via SLC35A2 in addition to UDP-Gal [39]. However, glycoproteins synthesized by the SLC35A2-deficient cell lines are very rich in terminal GalNAc residues as shown by staining with *Vicia villosa* lectin [137]. This is yet another example where the substrate specificity determined for an NST in a heterologous system does not seem to correspond to its native function.

There is one more controversy associated with SLC35A2. Despite the fact that it is the only so far identified UDP-Gal transporter in mammals, the MDCK-RCA^r^ mutant deficient in its activity was shown to synthesize normal amounts of chondroitin sulphate (CS) proteoglycans [136], although the tetra-saccharide linker through which CS chains are linked to core proteins contain two Gal residues. This suggests an existence of alternative to SLC35A2 UDP-Gal transporter.

Among the NSTs for which their three-dimensional structure has been determined, is the CMP-Sia transporter from *Z. mays* [106]. However, because plants do not incorporate Sia in their glycans [166], the presence of a CMP-Sia transporter in maize is difficult to rationalize.

Another potential controversy is related with UDP-Xyl supply for glycosylation. The need for transport of this NS into the Golgi is debatable, as UDP-Xyl was shown to be synthesized in the ER/Golgi lumen (and not in the cytoplasm) from UDP-GlcA due to the action of UDP-Xyl synthase (UXS) [167]. Hence, the presence of a dedicated UDP-Xyl transporter in the ER/Golgi membranes seems to be redundant.

For convenience, most important experimental observations (alongside with references) that add to the complexity of the NSTs world and that are not directly in line with the initial general concepts were summarized in Table 2. General tabularized information covering a larger repertoire of NSTs can be found in the review articles [168,169].

## 5. Closing Remarks

Starting from the 1970s, a substantial progress in the field of nucleotide transport has been accomplished, however, a picture of glycosylation in mammalian cells is still far from being clear. The results of our studies cast multiple doubts on the so far postulated idea of NS delivery to the mammalian Golgi. It becomes increasingly evident that there are separate routes in the cell for supply of the same NS to different glycosylation pathways.

This concept appears to be supported by the Golgi transport of UDP-GlcNAc. In our view, UDP-GlcNAc is selectively delivered to the branching Mgats (Mgat4 and Mgat5) by SLC35A3, while Mgat1 and Mgat2 seem to obtain their substrate from a different source. Moreover, although the SLC35A3 activity is indispensable for KS biosynthesis, it does not seem to support the production of another GlcNAc-rich GAG, i.e., HS, which in turn appears to depend on SLC35D2. Such selectivity could result from an existence of discrete spatially separated and functionally independent complexes consisting of unique sets of GTs and NSTs in the Golgi membranes. 

Our results provided a framework to continue studies on heterologous complexes formed by NSTs and related proteins. Precise mapping of such complexes would allow to better understand the mechanisms that govern and regulate the process of glycosylation. We believe this could be achieved using complementary techniques such as co-immunoprecipitation and proximity-based assays including those performed in living cells (FLIM-FRET, NanoBiT).

Based on our findings obtained in knockout-based studies, it can be suggested that many (if not all) NS are supplied to the Golgi by more than one transport route. It should be emphasized that many members of the SLC35 family were not assigned to specific substrates. Therefore, it may be reasonable to inactivate more than one SLC35 gene at a time as an extension of the generation and phenotypic characterization of single knockouts.

Finally, our recent results suggest that there is a functional connection between the processes that were thought to be interdependent, i.e., NST function and cytoplasmic NS synthesis. Our idea that GDP-Fuc could exist as several independent pools that are supplied to the Golgi by different routes sets a new direction in research on NS and their transporters.

## Figures and Tables

**Figure 1 ijms-23-08648-f001:**
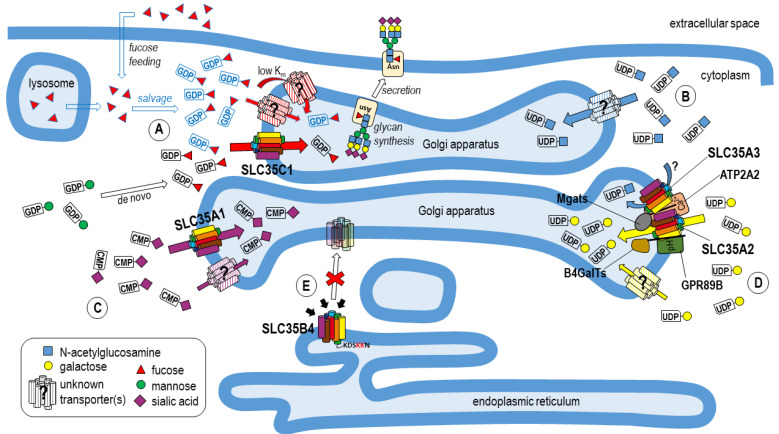
Translocation of selected NSs into the mammalian Golgi—a postulated view. (**A**) GDP-Fuc is transported in more than one route. SLC35C1 transporter carries the majority of the GDP-Fuc pool produced from GDP-Man in the de novo biosynthetic pathway but does not utilize the pool of the GDP-Fuc formed in the salvage biosynthetic pathway. The latter can be translocated by another carrier(s), including a potential one with a very low K_m_ able to utilize physiological GDP-Fuc concentrations. (**B**) The identity of the major transporter of UDP-GlcNAc is unclear. The involvement of SLC35A3 in the process of transport of UDP-GlcNAc for the formation of antennae in complex-type N-glycans cannot be excluded but it seems to be redundant for the wild-type glycosylation phenotype. (**C**) SLC35A1 is a major transporter of CMP-Sia but a minor portion of CMP-Sia can translocate to the Golgi in its absence through an unknown route. (**D**) UDP-Gal likely has an alternative transporter to SLC35A2. The latter forms a binary complex with SLC35A3 and both can interact with multiple other proteins including GlcNAc transferases (Mgats) and other proteins (ATPases e.g., ATP2A2 or pH regulators e.g., GPR89B, etc.). In addition, SLC35A2 associates with Gal transferases (B4GalTs), B4GalT1 and B4GalT4. (**E**) SLC35B4 is resident to the ER, not to the Golgi. Its correct localization is dependent on the presence of the native C-terminal sequence including a conserved dilysine motif. The perturbation of this motif including attachment of C-terminal tags results in protein mislocalization.

**Table 1 ijms-23-08648-t001:** A list of nucleotide sugars synthesized in mammalian cells and their commonly used abbreviations.

Nucleotide Sugar Full Name	Abbreviation
cytidine monophosphate sialic acid	CMP-Sia
guanosine diphosphate mannose	GDP-Man
guanosine diphosphate fucose	GDP-Fuc
uridine diphosphate galactose	UDP-Gal
uridine diphosphate glucose	UDP-Glc
uridine diphosphate glucuronic acid	UDP-GlcA
uridine diphosphate *N*-acetylglucosamine	UDP-GlcNAc
uridine diphosphate *N*-acetylgalactosamine	UDP-GalNAc
uridine diphosphate xylose	UDP-Xyl

**Table 2 ijms-23-08648-t002:** Molecular features, subcellular localization, specificity, discovery, and controversial observations concerning the NSTs described in this study. GA, Golgi apparatus; ER, endoplasmic reticulum.

NST Name	Molecular Features	Subcellular Localization	Substrate Specificity	Method of Discovery	Controversial Results/Observations
SLC35A1	36.8 kDa, 337 aa	GA	CMP-Sia	Gene cloning by complementation of the Lec2 mutant cell line [36]	- Sia is still incorporated into glycoconjugates produced by the HEK293T knockout cell line [98].
SLC35A2	41.0 kDa, 393 aa (UGT1), 41.3 kDa, 396 aa (UGT2)	GA (UGT1), ER/GA (UGT2)	UDP-Gal, UDP-GalNAc	Gene cloning by complementation of the Had-1 mutant cell line [33]	- Gal is still incorporated into glycoconjugates produced by the MDCK and CHO mutant cell lines [137] and HEK293T knockout cells [138] - Specificity towards UDP-GalNAc is not reflected by the phenotypes of the MDCK and CHO mutant cell lines [137]
SLC35A3	36.0 kDa, 325 aa	GA	UDP-GlcNAc	Gene cloning by complementation of the *K. lactis* mutant [34]	- No effect on GlcNAc incorporation into glycoconjugates produced by the CHO, HEK293T and HepG2 knockout cell lines [138]
SLC35C1	39.8 kDa, 364 aa	GA	GDP-Fuc	Gene cloning by complementation of the cells derived from LADII patients [11,37]	- Fuc is still incorporated into glycoconjugates produced by the HEK293T and HepG2 knockout cell lines [150] - Fucosylation in the HEK293T and HepG2 knockout cell lines is restored upon supplementation with exogenous fucose [150]
SLC35B4	37.4 kDa, 331 aa	GA/ER	UDP-Xyl, UDP-GlcNAc	Transport assay in a *Saccharomyces cerevisiae* heterologous system [63]	- Conflicting data regarding the subcellular localization (Golgi vs. ER) [64,144,155] - Lack of phenotypic effects in the HepG2 knockout cell line [145]

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
