# Peer review of "Delivery of Nucleotide Sugars to the Mammalian Golgi: A Very Well (un)Explained Story"

_ijms, 2022, doi:10.3390/ijms23158648_

Round 1
Reviewer 1 Report
In this review, the authors examine the field of nucleotide sugar transporters. The review nicely recaps what is known in the field, combining older data with more recent findings. The authors highlight interesting questions remaining in the field and provide a framework for future investigations.
Section 2.2
- The statement, “ generation of glycosylation-deficient clones by spontaneous or induced mutagenesis does not yet explain the genetic defect underlying the altered phenotypes” is confusing. Do the authors mean that glycosylation-deficient mutants created by spontaneous or induced mutagenesis does not identify the specific genetic defect that underlies the altered phenotype?
- The paragraph on the CHO-6B2 is confusing and needs to be reworded. Was the CHO-6B2 mutant shown to belong to the CHO-Lec2 complementation group by cell fusion? It may be better to have the first sentence state that “using ethylmethanesulfonate was used to generate the CHO-6B2 mutant that was found via cell fusion experiments to belong to the CHO-Lec2 complementation group.” Then, the complementation by the cDNA library can be combined with the next sentence…”The mutant was complemented with a murine cDNA library and the resulting ORF encoded a multi-transmembrane protein, by sequence analysis, that localized to the Golgi and showed high sequence similarity to plant ammonium transporters.”
Section 2.3
- For the summary, the authors state,”…proved the existence of carrier-mediated systems for which substrate specificity, Km, Vmas, temperature/inhibitor dependence, sensitivity to proteases/detergents, etc. could be measured.” Were any of these factors measured in the microsome penetration experiments? If so, could the authors write a brief summary of what was found?
Section 2.4
- It is unclear what the authors mean by, “transport was discriminative towards any of the tested UDP-NSs”.
Section 2.5
- The authors state, “ characteristics of UDP-Xyl transport in PC liposomes were perturbed as compared to those in Golgi lipids.” Could the authors summarize how they differed?
- For the CMP-Sia transporter reconstitution into artificial membranes, what were the results? Did the results agree with those found in the heterologous expression system in P. pastoris?
Section 3.1
- In the last sentence, the authors state that the crystal structures revealed the requirement for short-chain lipids in the membrane environment. Could the authors expand on this point and provide a little more detail?
Section 3.2
- The authors state that all of the NSTs identified so far localize either to the Golgi or the ER. The ER localization of UGT2 is dependent on a C-terminal motif. Do any of the other ER-localized NSTs have this motif?
- For the yeast GDP-Man transporter, the authors state that a sequence present in the N-terminal cytosolic tail was shown to be a determinant of its localization. Was a specific motif identified or was the N-terminal tail in general shown to be important?
Section 3.4
- Was the leucine zipper motif found in the yeast UDP-GlcNAc shown to be important for dimerization? If so, how? By mutational analysis?
- For Vrg4, the presence of lipids was revealed at the dimer interface but were the lipids shown to be necessary for dimer formation? If so, what technique was used?
Section 4.2
- If the SLC35A3 knockdown led to a depletion of tri- and tetraantennary species of complex-type N-glycans, why does the knockout only result in a mild decrease? Perhaps, the authors can discuss this point?
Section 4.3
- For the cell line CHO MAR-11, do they have the same phenotype as the CHO-Lec2 mutants? What is their phenotype?
Section 4.4
- In the 1999 Lübke study, did the authors identify that the genetic disorder LADII resulted in a decrease in fucosylation? The verb is missing from the sentence.
For the SLC35C1/SLC35C2 knockout, fucosylated glycans were still detectable. To the same level as in SLC35C only knockouts?
Author Response
Responces to Reviewer #1
In this review, the authors examine the field of nucleotide sugar transporters. The review nicely recaps what is known in the field, combining older data with more recent findings. The authors highlight interesting questions remaining in the field and provide a framework for future investigations.
We are grateful to the Reviewer #1 for this positive feedback to our efforts.
Section 2.2
- The statement, “ generation of glycosylation-deficient clones by spontaneous or induced mutagenesis does not yet explain the genetic defect underlying the altered phenotypes” is confusing. Do the authors mean that glycosylation-deficient mutants created by spontaneous or induced mutagenesis does not identify the specific genetic defect that underlies the altered phenotype?
We appreciate this specific suggestion. The respective fragment has been replaced with the clearer wording suggested by the Reviewer #1.
- The paragraph on the CHO-6B2 is confusing and needs to be reworded. Was the CHO-6B2 mutant shown to belong to the CHO-Lec2 complementation group by cell fusion? It may be better to have the first sentence state that “using ethylmethanesulfonate was used to generate the CHO-6B2 mutant that was found via cell fusion experiments to belong to the CHO-Lec2 complementation group.” Then, the complementation by the cDNA library can be combined with the next sentence…”The mutant was complemented with a murine cDNA library and the resulting ORF encoded a multi-transmembrane protein, by sequence analysis, that localized to the Golgi and showed high sequence similarity to plant ammonium transporters.”
We appreciate this specific suggestion. The respective fragment has been replaced with the clearer wording suggested by the Reviewer #1.
Section 2.3
- For the summary, the authors state,”…proved the existence of carrier-mediated systems for which substrate specificity, Km, Vmas, temperature/inhibitor dependence, sensitivity to proteases/detergents, etc. could be measured.” Were any of these factors measured in the microsome penetration experiments? If so, could the authors write a brief summary of what was found?
We agree with the Reviewer #1 that with respect to the detailing of the published outcomes, section 2.3 is far from being comprehensive but we would prefer to keep its current character. Chapter 2 was written with an intention to familiarize the reader with the principles of the techniques used in the field by referring to some selected pieces of research. More detailed information on the specific transporters/transport activities can be found in the specific references provided.
Section 2.4
- It is unclear what the authors mean by, “transport was discriminative towards any of the tested UDP-NSs”.
Agreed. The respective sentence has been rephrased.
Section 2.5
- The authors state, “ characteristics of UDP-Xyl transport in PC liposomes were perturbed as compared to those in Golgi lipids.” Could the authors summarize how they differed?
UDP-Xyl transport in PC liposomes was not temperature and inhibitor sensitive. This information has been added to the sentence.
- For the CMP-Sia transporter reconstitution into artificial membranes, what were the results? Did the results agree with those found in the heterologous expression system in P. pastoris?
The CMP-Sia transporters over-expressed in E. coli and in P. pastoris were handled and quality-checked in different manners which would require a longer description. To keep the focus of the section on reconstitution we have refrained from detailing other aspects.
Section 3.1
- In the last sentence, the authors state that the crystal structures revealed the requirement for short-chain lipids in the membrane environment. Could the authors expand on this point and provide a little more detail?
We provided some more details on this phenomenon in the revised version of the manuscript.
Section 3.2
- The authors state that all of the NSTs identified so far localize either to the Golgi or the ER. The ER localization of UGT2 is dependent on a C-terminal motif. Do any of the other ER-localized NSTs have this motif?
Yes, the C-terminal ER localization motif is present also in other NSTs, e.g. SLC35B1 (KKTSH), SLC35B4 (KDSKKN) and SLC35D1 (KGKGAV). The information has been added to the paragraph.
- For the yeast GDP-Man transporter, the authors state that a sequence present in the N-terminal cytosolic tail was shown to be a determinant of its localization. Was a specific motif identified or was the N-terminal tail in general shown to be important?
Amino acids 16-44 were shown to be important for the ER export of the GDP-Man transporter, but no specific sequence motif responsible for this phenomenon was identified. The corresponding fragment in the manuscript has been rephrased.
Section 3.4
- Was the leucine zipper motif found in the yeast UDP-GlcNAc shown to be important for dimerization? If so, how? By mutational analysis?
No studies aiming at demonstration of the involvement of the leucine zipper motif in dimerization of the yeast UDP-GlcNAc transporter were performed.
- For Vrg4, the presence of lipids was revealed at the dimer interface but were the lipids shown to be necessary for dimer formation? If so, what technique was used?
The authors of the work showed that lipids localize at the dimer interface (crystal structure, molecular dynamics) and that the presence of lipids increases thermal stability (thermal shift assay) and demonstrated that the dimer formation is dispensable for transport activity. No experiment testing the requirement of lipids for the dimer formation was presented.
Section 4.2
- If the SLC35A3 knockdown led to a depletion of tri- and tetraantennary species of complex-type N-glycans, why does the knockout only result in a mild decrease? Perhaps, the authors can discuss this point?
The word “mild’ was indeed a bit misleading and was removed from the sentence. In fact, the phenotypes of the knockdown and the knockout were very similar, i.e. a depletion of tri- and tetraantennary but not the biantennary N-glycans.
Section 4.3
- For the cell line CHO MAR-11, do they have the same phenotype as the CHO-Lec2 mutants? What is their phenotype?
MAR-11 phenotype is characterized by little surface sialic acid, even less than CHO-Le2. This information has been included in the sentence.
Section 4.4
- In the 1999 Lübke study, did the authors identify that the genetic disorder LADII resulted in a decrease in fucosylation? The verb is missing from the sentence.
The verb “associated” is present in the sentence: “In 1999 Lübke et al. associated genetic disorder…”
For the SLC35C1/SLC35C2 knockout, fucosylated glycans were still detectable. To the same level as in SLC35C only knockouts?
To the same level. The sentence was modified to include this information. The cited publication is already available at the following link https://pubmed.ncbi.nlm.nih.gov/35772493/. The corresponding citation has been updated.
Reviewer 2 Report
This review article is well and clearly written and the topic is relevant and important.
The review is written according to a historical perspective and includes very many details that make difficult for a reader to get to the specific points of interest. I strongly recommend the authors to prepare one or more tables categorizing transporters and/or their families, their main molecular and functional features, and the controversial aspects, together with the related references.
In addition, in a so wide dissertation there is not any mention about the biological and evolutionary meaning of nucleotide sugars transportation into the Golgi, taking into account the very many cytosolic glycosylations (including the ones operated by Golgi enzymes facing the cytosol, such as UGCG) that occur in eukaryotes despite the strong dilution of nucleotide sugars in the cytosol. I recommend filling these gaps in a revised version.
Author Response
Responces to Reviewer #2
This review article is well and clearly written and the topic is relevant and important.
We are grateful to the Reviewer #2 for this positive feedback to our efforts.
The review is written according to a historical perspective and includes very many details that make difficult for a reader to get to the specific points of interest. I strongly recommend the authors to prepare one or more tables categorizing transporters and/or their families, their main molecular and functional features, and the controversial aspects, together with the related references.
We are grateful to the Reviewer #2 for the suggestion. A table covering the most important information about the NSTs being the focus of our interest together with the related references has been included in the revised version of the manuscript (section 4.8). In addition, two citations to the related review articles that contain tabularized information on NSTs which might be useful to the reader have been appended.
In addition, in a so wide dissertation there is not any mention about the biological and evolutionary meaning of nucleotide sugars transportation into the Golgi, taking into account the very many cytosolic glycosylations (including the ones operated by Golgi enzymes facing the cytosol, such as UGCG) that occur in eukaryotes despite the strong dilution of nucleotide sugars in the cytosol. I recommend filling these gaps in a revised version.
We are grateful to the Reviewer #2 for the suggestion. A brief paragraph mentioning cytosolic glycosylation has been added to the section 1 of the revised manuscript.